# Are We Sure that Adjuvant Chemotherapy is the Best Approach for Resectable Pancreatic Cancer? Are We in the Era of Neoadjuvant Treatment? A Review of Current Literature

**DOI:** 10.3390/jcm8111922

**Published:** 2019-11-08

**Authors:** Ester Oneda, Alberto Zaniboni

**Affiliations:** Department of Clinical Oncology, Fondazione Poliambulanza, 25124 Brescia, Italy; ester.oneda@poliambulanza.it

**Keywords:** pancreatic cancer, adjuvant chemotherapy, neoadjuvant chemotherapy

## Abstract

The outcome of pancreatic cancer is poor, with a 9% 5-year survival rate. Current treatment recommendations in the 10%–20% of patients who present with resectable disease support upfront resection followed by adjuvant therapy. Until now, only early complete surgical (R0) resection and adjuvant chemotherapy (AC) with either FOLFIRINOX (5-fluorouracil, leucovorin, irinotecan, and oxaliplatin) or nab-paclitaxel plus gemcitabine have been shown to prolong the survival. However, up to 30% of patients do not receive adjuvant therapy because of the development of early recurrence, postoperative complications, comorbidities, and reduced performance status. The aims of neoadjuvant chemotherapy (NAC) are to identify rapidly progressing patients to avoid futile surgery, eliminate micrometastases, increase the feasibility of R0 resection, and ensure the completion of multimodal treatment. Neoadjuvant treatments are effective, but there is no consensus on their use in resectable pancreatic cancer (RPC) because of its lack of a survival benefit over adjuvant therapy. In this review, we analyze the advantages and disadvantages of the two therapeutic approaches in RPC. We need studies that compare the two approaches and can identify the appropriate sequence of adjuvant therapy after neoadjuvant treatment and surgery.

## 1. Introduction

Pancreatic cancer (PC) is one of the most deadly cancers in developed countries. The incidence of PC in western Europe is 7.3 per 100,000 individuals, and the mortality rate is 6.8 per 100,000 individuals [1]. PC represents the fourth leading cause of death related to cancer after lung, colorectal, and prostate cancers in men and after breast, colorectal and lung cancers in women [2,3], with a 5-year survival rate of 9% when considering all stages [4,5]. Pancreatic cancer mortality has been increasing in both genders in recent decades [1]; this increase could be related to the gain of therapeutic advantages in other types of cancers compared with PC and to the growing prevalence of pancreatic cancer [6].

For the 10%–20% of patients with resectable disease, clinical data support upfront resection and adjuvant therapy. In borderline resectable pancreatic cancer (BRPC), surgical resection could follow induction chemotherapy to reduce tumor size and control potential micrometastatic disease [7,8]. Although surgical techniques have advanced and adjuvant treatment has been standardized, survival rates remain low; early total surgical (R0) resection with lymph node dissection and removal of the extrapancreatic nerve plexus and celiac axis [9,10] are the key factors influencing survival. Adjuvant therapy prolonged survival after surgery and has become the standard treatment for resectable pancreatic cancer (RPC). Adjuvant chemotherapy (AC) with 5-fluorouracil plus folinic acid or gemcitabine has already been shown to prolong the 5-year survival by approximately 16%–21% [11,12,13], and the new combination regimens have another promising advantage. However, in clinical practice many patients develop early relapse despite complete surgical resection [14,15]. Approximately 30% of patients fail to receive adjuvant therapy because of postoperative complications, early metastases, reduced performance status, and comorbidities [16].

This lack of efficacy in patients allowed for the development of neoadjuvant chemotherapy (NAC) with a focus on identifying aggressive tumors in order to avoid inappropriate surgery, eliminate micrometastases, increase complete resectability (at the R0 stage), reduce tumor seeding risk at the time of surgery and complete multimodal treatments [7,8]. There are data suggesting that most PCs are likely to exhibit micrometastases even when only a small primary tumor is detectable by clinical imaging, leading to early relapse and death after surgery. Effective chemotherapy could treat these micrometastases and prevent early post-surgery recurrence [17]. Possible disadvantages of NAC are the potential complications associated with biliary stents or the necessity for biliary decompression during treatment and the delay of surgery. These events could lead to disease progression (in approximately 20% of PCs [18,19]), a potential increase in the occurrence of postoperative complications [20] and only a few benefits in overall survival (OS) and disease-free survival (DFS), as has been observed in several meta-analyses [21,22]. Despite this, NAC is widely accepted for the management of locally advanced (LA) and borderline resectable (BR) cases [7,8,23], but no evidence exists on the benefit in RPC.

Neither randomized trials nor reviews identify a survival advantage of neoadjuvant treatment because they include different patient types (BR, LA, and RPC), the number of patients is often insufficient, the surgery was performed in a low-volume surgery hospital, resectability criteria and the criteria to proceed after neoadjuvant treatment lack consensus, and the mOS is often underestimated because many patients were not eligible to receive curative resection. Furthermore, it is not clear whether any adjuvant regimen should be delivered after neoadjuvant treatment and surgery and which regimen should be used.

## 2. Neodjuvant Treatment—The Data

Ideally, neoadjuvant chemotherapy could be offered to a greater number of patients than adjuvant chemotherapy because of its early administration in patients with a better performance status than patients who have undergone resection, which allows them to have good tolerance to multi-agent regimens and a higher negative-margin resection rate than that seen after upfront surgery. Randomized trials have tried to show which is the most suitable neoadjuvant treatment.

Tajima et al. [20] treated 52 BR patients with neoadjuvant therapy with GEM-based regimens (GEM monotherapy, GEM plus S-1 therapy, or nabPTX plus GEM therapy) for two cycles, and then surgery was performed. In the control group, 34 patients underwent upfront surgery. Both arms received adjuvant gemcitabine after resection. Partial response (PR) was achieved in five of the 52 NAC group patients, stable disease (SD) was achieved in 45 patients (86.5%), and progressive disease (PD) occurred in 2 patients (3.9%); however, on computed tomography (CT) scan, no significant tumor volumetric reduction was observed after NAC, and only the CA 19.9 mean value significantly decreased. All tumor samples showed histopathological cell tumor injury, although pathological complete response (CR) was not seen in any patient. Unfortunately, the 5-year OS rates in the NAC group compared with the control group were not increased. A total of 40.7% of patients relapsed within one year after surgery. The negative predictors of response were the CA 19.9 value, tumor stage, number of lymph node metastases, and rates of nerve and plexus invasion [20].

The phase II/III trial JSAP05 from Japan showed that in BR and resectable cancer patients, there was a resection rate of 93% and an mOS of 36.72 months with neoadjuvant treatment with the combination of gemcitabine plus S1, an orally active fluoropyrimidine, followed by surgery and adjuvant treatment with S1, as compared to a resection rate of 82% and an mOS of 26.65 months seen with upfront surgery (US) plus adjuvant treatment with S1 [24]. Furthermore, neoadjuvant treatment improved the 2-year OS rate from 52.5% to 63.7%. The benefit from neoadjuvant treatment was observed in most subgroups in the forest plot analysis. The time to surgery and surgery-related morbidity was not modified with neoadjuvant gemcitabine plus S1 treatment. Additionally, pN1 occurrence was significantly lower in the NAC arm than in the US arm (59.6% vs. 81.5%) [24]. For a different metabolism of the drug, the Western population showed higher toxicity with S1 than Asians, and studies are needed to evaluate the efficacy of S1 in that population. In addition, other studies have been carried out in Western population; Janssen et al. [25] analyzed the effects of neoadjuvant FOLFIRINOX (5-fluorouracil, leucovorin, irinotecan, and oxaliplatin) in patients with BRPC and LAPC and found a good resection rate (67.8%; 95% CI: 60.1–74.6), while the R0 resection rate was 83.9% (95% CI: 76.8–89.1). The median OS was 22.2 months (95% CI: 18.8–25.6), and the median PFS was 18.0 months (95% CI: 14.5–21.5) [25]. Because FOLFIRINOX was more effective than gemcitabine alone in patients with advanced pancreatic cancer [26], it is expected that it could improve OS in a neoadjuvant setting. Recently, Michelakos et al. published data from Massachusetts General Hospital showing a median OS of 37.7 months after neoadjuvant FOLFIRINOX [27].

PREOPANC-1, a phase III randomized trial comparing neoadjuvant chemoradiotherapy (NACRT) followed by surgery, found a better DFS with NACRT than that seen in patients who had upfront surgery (US). Furthermore, the 2-year survival rate was higher for NARCT than for US (42% vs. 30%). The study enrolled 246 patients with BRPC, who were randomly assigned to receive US or chemoradiotherapy (15 fractions of 2.4 gray (Gy) combined with weekly gemcitabine) followed by surgery. Both groups also received AC with gemcitabine. The median overall survival was 17.1 months with NARCT compared to 13.7 months with US. The time until cancer recurrence was longer with preoperative therapy (9.9 vs. 7.9 months) than without preoperative therapy. Among the resected patients, the difference in median survival was even greater: 42.1 months with preoperative treatment vs. 16.8 months with US. The resection rate was 72% in the US group and 62% in the chemoradiotherapy group. Among the resected patients, the tumor was microscopically completely removed in the 63% of patients who received NACRT and in 31% of patients who receive US [28]. In addition, patients treated with NACRT had less surgical site infections (SSIs), a higher curative resection rate, and an earlier stage of disease than patients who were not treated. On the other hand, an important increase of the risk of mortality is a low muscle mass at diagnosis, associated with loss of lean tissue during chemotherapy. Sarcopenia was prevalent in half of patients at the time of diagnosis with PC. NACRT can induce a low prognostic nutritional index (PNI), that can be associated with the poor patient condition and low tolerability of adjuvant chemotherapy [29,30,31]. Recent guidelines recommended assessing the patient’s nutritional status from the start of treatment and monitoring it [30]. With these considerations, NAC should be avoided in sarcopenic patients to give them a chance at resection. Future clinical trials comparing neoadjuvant chemoradiation with neoadjuvant chemotherapy are needed to confirm these results.

The phase II SWOG S1505 (ClinicalTrials.gov, NCT02562716) trial is investigating the safety and effectiveness of ABRAXANE in combination with gemcitabine in neoadjuvant environments. Preliminary data show that (42; 29%) patients were ineligible for the study because of venous involvement ≥ 180°, arterial involvement, or distant disease at the time of diagnosis [32,33]. Of the eligible patients, 73% underwent resection. We do not yet have data on the number of patients completing the treatment and on DFS and OS.

ESPAC-5: A multi-centre, prospective, randomized phase II trial, is recruiting patients to compare neoadjuvant therapy (GemCap or FOLFIRINOX) or chemoradiotherapy to immediate surgical exploration in patients with borderline resectable pancreatic cancer. All resected patients will also receive adjuvant chemotherapy with either gemcitabine or 5-fluorouracil for six cycles according to physician choice [34]. As can be seen in Table 1 and Table 2 (with small retrospective studies), it is not possible to make a direct comparison between the studies and reviews listed.

## 3. Adjuvant Treatment—The Data

Up to a 30% 5-year survival rate can be achieved if adjuvant chemotherapy is delivered after surgical resection [52]. Since CONKO-001 demonstrated the efficacy of gemcitabine (GEM) as a postoperative adjuvant treatment in 2007 [13,53] with a 5-year survival rate of 20.7%, it has become the standard of care.

The 5-year survival rate was estimated in the ESPAC-1, ESPAC-3, and ESPAC-4 trials (fluorouracil vs. control, gemcitabine vs. fluorouracil, gemcitabine plus capecitabine vs. gemcitabine); it was 21% vs. 8%, 17.5% vs. 15.9%, and 28.8% vs. 16.3% respectively [11,52,54]. ESPAC-3 did not show a survival benefit of adjuvant gemcitabine than 5-fluoruracil and folinic acid [54]. In ESPAC-4 [52], the combination of gemcitabine and capecitabine (an orally active prodrug of 5-fluoruracil) performed better than gemcitabine with an increase in progression-free survival and overall survival 28 vs. 25.5 months (HR 0.82 (95% CI 0.68–0.98), *p* = 0.032). One factor correlates to survival is the resection margins status, the median overall survival was 23.7 vs. 23 months for patients who had positive resection margins (R1 status) and 39.5 vs. 27.9 months in who had negative resection margins (R0 status). Other factors that emerged as related to survival were smoking, preoperative, and postoperative CA19.9 serum levels, preoperative C-reactive protein concentrations, tumor grade, lymph nodes status, maximum tumor size, tumor stage, venous resection, and local invasion.

The JASPAC-1 trial also showed superior survival with S1 compared to gemcitabine. The 5-year OS was 44.1% in the S-1 group and 24.4% in the gemcitabine group. Tests of S1 are required to assess its efficacy and the safety in Western populations [55].

The recent trial PRODIGE showed that combination chemotherapy with fluorouracil, leucovorin, irinotecan, and oxaliplatin (FOLFIRINOX), previously tested in the metastatic setting [26], lead to longer disease-free survival, overall survival, metastasis-free survival, and cancer-specific survival than gemcitabine therapy [56]. The median overall survival was 54.4 months in the modified-FOLFIRINOX group and 35 months in the gemcitabine group. The 3-years OS rate was 63.4% in the modified-FOLFIRINOX group and 48.6% in the gemcitabine group. As expected, the modified FOLFIRINOX regimen was more toxic than gemcitabine; in particular, grade 3 or 4 neutropenia and diarrhea occurred in 75.9% of the patients in the modified-FOLFIRINOX group and in 52.9% of patients in the gemcitabine group [56].

A recent report from the phase III APACT study showed that the use of adjuvant treatment with nab-paclitaxel (ABRAXANE) in combination with gemcitabine did not improve disease-free survival compared to the use of gemcitabine alone (mDFS 19.4 months vs. 18.8 months), while overall survival was improved (mOS 40.5 months vs. 36.2 months) [57]. The results of these studies are resumed in Table 3.

Valle et al. reported that an independent prognostic factor after PC resection was the completion of planned cycles of adjuvant chemotherapy, rather than its early administration [37]. In many trials, with highly selected patients who have completely recovered from surgery, the completion rate of chemotherapy ranges from 54 to 79% [26,52]. A.M. Altman et al. [58] studied the rate of completion of AC and the factors associated with completion. They collected 2440 patients who underwent upfront surgery, 65% of the patients received no AC, 28% received incomplete AC, and only 7% completed six cycle of AC [58]. T. Akahori et al. [59] analyzed 135 patients: 90 patients completed planned adjuvant chemotherapy, while 45 patients failed to complete adjuvant chemotherapy. A total of 14.8% of patients experience recurrence during the treatment, and 18.5% of the patients ceased treatment before completion because of poor performance status.

In conclusion, the factors associated with the completion of chemotherapy after resection are performance status and disease-related comorbidities at the time of diagnosis [60,61,62] and surgical morbidity; half of the patients present with a major postoperative complication [61,63,64]. Indeed, a serum CA 19.9 level > 85 U/mL within 6 months after upfront surgery was an independent risk factor for recurrence [20,29]. Predictors of poor prognosis associated with failure to complete adjuvant chemotherapy included early recurrence, high serum CA 19.9, presence of lymph node metastasis, and positive surgical margins [65,66], low preoperative prognostic nutritional index, intraoperative blood transfusion, infection of the surgical site, and advanced tumor stage. Studies are needed to find the most effective and least toxic chemotherapy regimen to overcome these problems.

## 4. Discussion

The only way to cure PC is R0 resection, but R1 resections are reported in many cases [67], and even after undergoing curative resection, the local recurrence rate is 50%–80% and the probability of developing distant metastases is 25%–50% [68]. Multiple trials have shown a survival benefit for adjuvant chemotherapy after resection of PC, but not all RPC patients are able to complete the planned AC because of early recurrence during therapy (approximately 34% [63,69]) or to poor postoperative patient condition. Neoadjuvant treatment could improve the R0 resection rate, reduce the risk of early metastasis, and consequently increase overall survival. Short induction chemotherapy followed by consolidation therapy after resection has been shown to be successful in other solid gastrointestinal tumors, but it has not been observed in PC. The median OS seems to be inferior with neoadjuvant treatment compared with adjuvant treatment in many metanalyses; the mOS with adjuvant FOLFIRINOX chemotherapy in the PRODIGE24 trial [56] was 54.4 months while in the neoadjuvant setting, it is approximately 24 months [25]. The patients in the adjuvant and neoadjuvant trials were very different and cannot be compared directly. In fact, for adjuvant trials, the patients are required to meet certain criteria: complete macroscopic resection, complete recovery after surgery, and level of CA 19.9 under stabilized values; many patients are not included because of surgical complications and metastases that were found during surgery. Only one-third of patients receive adjuvant therapy.

No study has been designed to compare the survival in RPC with neoadjuvant vs. adjuvant chemotherapy. The still unanswered question is whether NAT or upfront surgery and adjuvant therapy are the appropriate strategies in resectable pancreatic cancer. Until now, the published studies included a mixture of patients with resectable or borderline resectable or even unresectable tumors, so we cannot determine clear conclusions, we urgently need clarification.

Nowadays several trials are recruiting resectable pancreatic cancer patients worldwide and are comparing the systemic neoadjuvant and adjuvant treatments. In Germany, the NEONAX trial is comparing 2 cycles of nab-paclitaxel (125 mg/m^2^)/gemcitabine (1000 mg/m^2^, on d1, 8 and 15 of a 28-day cycle) followed by surgery and 4 cycles of nab-paclitaxel/gemcitabine therapy, with surgery followed by 6 cycles of nab-paclitaxel/gemcitabine [70]. The rational of the use of two cycles of NAT take into consideration the lower tolerability with increasing therapy cycles and the good effectiveness (tumor regression in 30% of patients in a phase I trial) [71]. Furthermore, in a phase I/II trial, a reduction in the absorption of FDG was observed after 6 weeks of treatment with nab-paclitaxel/gemcitabine, suggesting that 2 cycles of this regimen are sufficient [72], and a further delay of surgery could reduce patient compliance. Other trials are recruiting patients: the NEOPAC study (NCT01314027, comparing neoadjuvant gemcitabine and oxaliplatin regimens with upfront surgery, all followed by adjuvant gemcitabine) [73]; the NEPAFOX trial (NCT02172976, comparing neoadjuvant chemotherapy with FOLFIRINOX, US follow by adjuvant FOLFIRINOX, and US follow by adjuvant gemcitabine) [74]; NorPACT-1 trial (NCT02919787, comparing neoadjuvant FOLFIRINOX with upfront surgery; both groups receive adjuvant chemotherapy gemcitabine and capecitabine) [75]; the PANACHE01-PRODIGE48 trial (NCT02959879, comparing two regimens of neoadjuvant chemotherapy: 4 cycles of mFOLFIRINOX or 4 cycles of FOLFOX; both are followed by surgery and adjuvant therapy) [76]; and the PREOPANC-2 trial (NTR7292, comparing neoadjuvant chemotherapy with FOLFIRINOX or gemcitabine plus capecitabine, with neoadjuvant gemcitabine-based CRT with upfront surgery). We are waiting for the results (studies are resumed in Table 4).

Subsequently, the problem of which treatment to propose after relapse of the disease will arise. Gbolahan et al. [77] observed 10-month median post-relapse OS (mOS) in patients with resected pancreatic cancer who had received combination therapy or monotherapy after relapse and 3-month mOS in patients undergoing best supportive care (BSC). In particular, the 14-month OS improvement was much more evident with a standard combination therapy regimen than with a single agent or non-standard combination chemotherapy [77]. There is a lack of studies on the use of chemotherapy in patients with resected PC who relapsed after an initial curative therapy with both neoadjuvant and adjuvant treatment, and it is not clear to what extent the first therapy may influence the efficacy and tolerability of the subsequent chemotherapy.

## 5. Conclusions

The outcome of pancreatic cancer remains poor, but survival is slowly increasing with a reported median OS of 54.4 months with adjuvant chemotherapy with FOLFIRINOX after surgery [56] while, until now, there has been no evidence of a survival benefit with neoadjuvant treatment in resectable pancreatic cancer patients. In order to improve the survival, the sequence of neoadjuvant and adjuvant treatment need to be explored in clinical trials. Clinicians are awaiting guidance on which therapeutic strategy is the best. The patients and regimens used in recent studies are very different and cannot be compared directly; we are awaiting the results of ongoing trials, which will hopefully clarify the current state of the field.

## Figures and Tables

**Table 1 jcm-08-01922-t001:** Analysis of survival and R0 status with neoadjuvant treatments according to updated studies and reviews.

	JSAP-05 [24]		Tajima [20]		Michelakos [27]		PREOPANC 1 [28]		Jassen [25]	
Type of pz (*n*)	RPC, BRPC (182)	RPC, BRPC (180)	RPC		BRCP (69), LAPC (71)		BRCP (246)		BRCP (283)	LAPC (315)
Regimen	Gem + S1 + surgery + S1 (adj)	Surgery + S1 (adj)	Gem regimen (gem+S1/gem/nab-P+gem) + surgery + gem (adj)	Surgery + gem (adj)	FOLFIRINOX + surgery	Surgery	NACRT + surgery + gem (adj)	Surgery + gem (adj)	FOLFIRINOX + surgery	FOLFIRINOX + surgery
mOS (months)	36.72	26.65	41.6	23.3	37.7	25.1	17.1	13.7	22.2	24.2
HR							0.74			
mDFS (months)					29.1	13.7	9.9	7.9	18	
HR							0.23			
R0 (%)			80.8	79.04	78 (R tot)		63	31	83.9	27

BRPC = borderline resectable pancreatic cancer; LAPC = locally advance pancreatic cancer; RPC = resectable pancreatic cancer; Gem = gemcitabine; nab-P = nab-paclitaxel; FOLFIRINOX = combination chemotherapy with fluorouracil, leucovorin, irinotecan, and oxaliplatin; NACRT= neoadjuvant chemoradiotherapy with weekly gemcitabine.

**Table 2 jcm-08-01922-t002:** Summary of prospective and retrospective trials to assess the efficacy of neoadjuvant treatments (not cited in the text).

Study	Type of pz (*n*)	Regimen (*n*)	mOS (Month)	PD (*n*)	PR (*n*)	SD (*n*)	Resection (*n*)
Kunzmann V et al. [35]	8 LAPC	NabP + gemFOLFIRINOX (adj)			5	3	3
Sueyoshi H et al. [36]	14 LAPC	W-nabP +gem + RT		4	2	6	
Reni M et al. [37]	24 LAPC	NabP + cis/cape/gem (PAXG)			16	8	6 (3 R0)
Reni M et al. [38]	223 LAPC	NabP + gem (28) or others gem regimens (195)		11	106	103	61 (38 R0)
LAPACT [39]	107 LAPC	NabP + gem		5	35	83	16 (7 R0)
PACT-19 [40]	54 (BRPC and LAPC)	NabP + gem (28) or nabP + cis/cape/gem (PAXG) (26)					17
Khushman M et al. [41]	51 LAPC	FOLFIRINOX + CRT					10 R0
Nitsche U et al. [42]	14 LAPC	FOLFIRINOX		1	6	6	4
Hackert T et al. [43]	575 LAPC	FOLFIRINOX (125) or gem regimens (322) or others regimens (128)	16				76 (31 R0)
Pfeiffer P et al. [44]	59 (BRPC and LAPC)	FOLFIRINOX + CRT					16
Suker M et al. [45]	315	FOLFIRINOX					81 (63 R0)
Lloyd S et al. [46]	115 (BRPC and LAPC)	CRT or CT or CCRT	12.5, 13.9, 21.5				53 (2 R0)
Casadei R et al. [47]	18 LAPC	CRT	22.4	5	4	18	7 R0
Fujii et al. [48]	21 BRPC	CRT	29.1				18
Fujii et al. [49]	504 BRPC and RPC	CRT	28.6				
Satoi S at al. [50]	67(BRPC and LAPC)	CRT+S1 (35) or gem + S1 (32)	22				31, 25
Murakami Y et al. [51]	52 RPC	Gem + S1	27.1				38

BRPC = borderline resectable pancreatic cancer; LAPC = locally advance pancreatic cancer; RPC = resectable pancreatic cancer; CCRT = chemotherapy followed by chemoradiation therapy; CRT = chemoradiotherapy; CT = chemotherapy; Gem= gemcitabine; nabP = nab-paclitaxel; mFOLFIRINOX = combination chemotherapy with fluorouracil, leucovorin, irinotecan, and oxaliplatin; Cis = cisplatin; Cape = capecitabine; PD = progressive disease; PR = partial response; SD = stable disease.

**Table 3 jcm-08-01922-t003:** Survival with selected adjuvant treatment in resected pancreatic cancer.

	CONKO-001		ESPAC-1		ESPAC-3v		ESPAC-4		JASPAC-1		PRODIGE 24		APACT	
Regimen	Gem	Obs	5 FU + Folinic Acid	Obs	5 FU + Folinic Acid	Gem	Gem + Cape	Gem	S1	Gem	mFOLFIRINOX	Gem	PacliT + Gem	Gem
mOS (months)	22.8	20.2	20.1	15.5	23.1	23.6	28	25.5	46.5	25.5	54.4	35	40.5	36.2
HR (95% CI)	0.76		0.66		0.94		0.82		0.57		0.64		0.82	
mDFS (months)	13.4	6.7			14.1	14.3	13.9	13.1	22.9	11.3	21.6	12.8	19.4	18.8
HR (95% CI)	0.55				0.96		0.86		0.60		0.58		0.88	
5 yrs OS (%)	20.7	10.4	21.1	8	15.9	17.5	28.8	16.3	44.1	24.4	63.4 (3 yrs OS)	48.6 (3 yrs OS)		

Obs = observation; Gem = gemcitabine; 5 FU = 5-fluorouracil; Cape = capecitabine; mFOLFIRINOX = combination chemotherapy with fluorouracil, leucovorin, irinotecan, and oxaliplatin.

**Table 4 jcm-08-01922-t004:** Ongoing trials.

	Arm A	Arm B	Arm C
NEONAX	Neoadjuvant nab-P+gem + surgery + adjuvant nab-P + Gem	Surgery + adjuvant nab-P + Gem	
NEOPAC	Neoadjuvant gem + surgery+ adjuvant gem	Neoadjuvant Oxaliplatin + surgery + adjuvant gem	Surgery + adjuvant gem
NEPAFOX	Neoadjuvant mFOLFIRINOX	Adjuvant mFOLFIRINOX	Adjuvant gem
NorPACT-1	Neoadjuvant mFOLFIRINOX + surgery + adjuvant gem plus cape	Surgery + adjuvant gemcitabine plus capecitabine	
PANACHE01-PRODIGE48	Neoadjuvant mFOLFIRINOX + surgery + adjuvant therapy	Neoadjuvant FOLOX + surgery + adjuvant therapy	
PREOPANC-2	Neoadjuvant mFOLFIRINOX	Neoadjuvant gem-based chemoradiotherapy	
ESPAC-5	Neoadjuvant mFOLFIRINOX or Gem plus cape	Neoadjuvant chemoradiotherapy	Up front surgery

Gem = gemcitabine; nab-P = nab-paclitaxel; mFOLFIRINOX = combination chemotherapy with fluorouracil, leucovorin, irinotecan, and oxaliplatin; FOLFOX = combination chemotherapy with fluorouracil, leucovorin and oxaliplatin.

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
