# Peer review of "Are We Sure that Adjuvant Chemotherapy is the Best Approach for Resectable Pancreatic Cancer? Are We in the Era of Neoadjuvant Treatment? A Review of Current Literature"

_jcm, 2019, doi:10.3390/jcm8111922_

Round 1

Reviewer 1 Report

This is a review on adjuvant or neoadjuvant therapy in pancreatic cancer. The review is not systematic, selective and really more like a personal view. I do not think that the overall conclusions are based well on the papers cited. The authors should ´try to work in more of current evidence and take out far reaching conclusions that are not based on evidence.

I have the following major concerns:

Already the title is misleading. For resectable pancreatic cancer, upfront resection and adjuvant therapy is the state of the art (as the authors write in the body of the text). While the concept of neoadjuvant therapy (NAT) is tested in resectable pancreatic cancer new do NOT live in an era of NAT!

The authors should give a better overview on evidence of NAT vs upfront resection and adjuvant therapy. There are more small RCTs and there are retrospective comparative data. The problem in the latter is selection and immortality bias that do not allow fair comparisons. RCTs with ITT analysis are required. The authors only provide a table on ongoing trials, which is not very helpful because it does not provide any answer. The authors should provide a table that lists the currently available data. And in the text they should address the problems of these data. The scientifically correct interpretation of all data at this moment is that the question NAT or upfront surgery and adjuvant therapy remains unanswered!   The problem in the literature is also, that studies include a mixture of patients with resectable or borderline resectable or even unresectable tumors. This should be discussed in more detail and included in a new table (see comment 2).   In the abstract it sounds as 5FU or Gem would be state of the art in adjuvant therapy. The authoprs only refer to 5 year survival. They should state the current best therapies (gemCAP/Folfirinox) here, as they do in the text.

The manuscript contains multiple grammatical mistakes and wrong expressions and should be proofred by a native speaker.   Some citations are wrong. E.g. ref 9/10 are not about removal of vessels…

Author Response

Thank you very much for your precious advice.
We know that the current state of art in resected pancreatic cancer is surgery plus adjuvant therapy. Since the Era of neoadjuvant therapy in this cancer type hasn’t come yet differently to other tumors, we wanted to arouse a stimulus in order to encourage new studies which will compare adjuvant and neoajduvant treatment in resected pancreatic cancer patients. We hope that the new title could be the correct compromise between our point of view and the current one.
Until now the studies on neoadjuvant chemotherapy lack of direct comparable data with adjuvant one. We have tried to retrieve the current data that compare neoadjuvant treatment to upfront surgery in two new tables, highlighting how the parameters and criteria for selecting patients are different from study to study.
We apologize for the quote in the abstract and therefore we provide a correction as you suggested.
We are providing an English revision to ensure grammatical correct form.

I wish your advice could give the expected outcome.

Reviewer 2 Report

Is “mOS” and “mDFS” mentioned several times within the manuscript and in Table 1 for median survival? Please clarify.

Please pay attention to: P = .023 (line 102) and p=0·032 (line 141).

Please clarify “PRC” in Table 1 heading.

Headings of Tables 1 and 2 should be more informative.

Cumulative presentation of RCTs for neoadjuvant and adjuvant chemotherapy and their results as Tables might be useful for the readers.

Author Response

Thanks for the revision.
We have tried to clarify the acronyms and better explain the contents of tables and headings.

We have also added two new tables to resume the current information of neoadjuvant chemotherapy trials.

Hoping for a better feedback, best regards.

Round 2

Reviewer 1 Report

The authors have adequuately addressed my concerns raised in the first Review and I think the manuscript gives now a more objective message and has been significantly improved.

I only have minor comments:

In the Abstract the authors mention FOLFOXIRI. This must be a mistake and I assume they wanted to menbtion FOLFIRINOX. Please revise/clarify and check throughout the manuscript. Another spelling mistake in the conclusions., p8263: "explore" Needs to be "explored".

Author Response

Thank you, we are pleased from your reply and thak you for updated the corrections. 

You are right, we are correcting these mistakes. 

Best regards